# A Metacoupling Framework for Exploring Transboundary Watershed Management

**Leandra Merz** [1,*] , **Di Yang** [2] **and Vanessa Hull** [3]

1   Geography Department, University of Florida, Gainesville, FL 32611, USA
2   Spatial Analysis Lab, University of Montana, Missoula, MT 59812, USA; di.yang@mso.umt.edu
3   Department of Wildlife Ecology and Conservation, University of Florida, Gainesville, FL 32611, USA; Vhull@ufl.edu
*   Correspondence: Leandra6@ufl.edu

**Abstract:** Water is crucial for ecosystem health and socioeconomic development, but water scarcity is becoming a global concern. Management of transboundary watersheds is inherently challenging and has the potential to lead to conflict over the allocation of water resources. The metacoupling framework, which explores the relationships between coupled human and natural systems that are nested within multiple different scales, has been proposed to inform more holistic management of transboundary watersheds. This paper provides the first attempt to apply a metacoupling framework to a transboundary watershed for an improved integrated understanding of this complex system at multiple spatial scales. It does so with the transnational Limpopo River watershed in Southern Africa, which covers 1.3% of the continent and supports the livelihoods of 18.8 million people living in Botswana, Mozambique, South Africa, and Zimbabwe. Sub-Saharan Africa is experiencing a growing gap between water availability and demand; the primary drivers are population growth and agriculture expansion. The novelty of the paper is outlining the importance of applying a metacoupling framework to transboundary watersheds, identifying the limitations to this application, and providing a detailed assessment of the steps needed to complete this application. We also identify directions for future research including application of a metacoupling framework to other transboundary watersheds and exploration of spillover effects and externalities within this and other transboundary watersheds.

**Keywords:** metacoupling; telecoupling; Limpopo River watershed; transboundary watersheds; coupled human and natural systems

## 1. Introduction

Water is one of the most essential elements for both natural systems and for socioeconomic development [1]. Yet around the world, one in three freshwater species is threatened with extinction [2] and an estimated 2.1 billion people lack access to clean drinking water as of 2015 [3]. The global water crisis is ranked as one of the top five threats to society and is predicted to become the number one threat to society over the coming decade [4]. Water scarcity, a combination of availability and quality, is becoming prevalent even in areas of high rainfall [5]. As the global human population continues to increase rapidly, there is an exponential increase in the demand for freshwater resources, particularly for agricultural irrigation [1,6]. It is estimated that 70% of global freshwater used is for agricultural purposes [7]. The land is rapidly being converted to agriculture and agricultural lands are increasingly being put under irrigation often without formal water management practices. This has resulted in tributaries drying up throughout the world [8].

These increasing demands and mounting threats are but one of the challenges faced by watershed management systems [9]. Among the most important of these is globalization. As distant places around the world become increasingly connected through trade and travel, water sources experience increased stress from distant sources and increased challenges in identifying and managing diverse anthropogenic threats spread out across space [9]. This is exemplified by examining the case of transboundary watersheds, large watersheds that span two or more countries and incorporate varied and often competing uses. There are 263 transboundary river basins that cover 45.3% of the land surface of the earth and support approximately 40% of the global human population [10]. Transboundary watershed management requires the cooperation of at least two nations and is heavily influenced by politics, economics, demographics, physical geography, and ecology among other factors [10,11]. There is great potential for conflict over the management of transboundary water resources and conflicts have already occurred in some of these basins [12].

Several frameworks that attempt to improve watershed management by considering both physical characteristics and the broader socioeconomic context have been proposed. For instance, hydro economic modeling (HEM) combines physical, environmental, economic, and engineering aspects of a water system into one framework to maximize the benefits of water allocation [13]. This framework focuses on the economic valuation of water to diagnose challenges in managing water resources and to improve allocation [14]. HEMs are noted for their ability to predict future scenarios and can even be applied to watersheds with limited data available [15,16]. While it can provide managers opportunities for improving the efficiency and transparency of water management, it fails to account for critical social aspects such as cultural values, social networks, and governance styles. The failure to address these important social aspects affecting water demand leaves the framework unable to properly manage water for all needs and in all contexts.

An alternative to HEM is socio hydrology (SH), which integrates both social and hydrological components to improve management of water resources [17]. It allows for the recognition of non-market values for water such as identity, well-being, spirituality, and social exchange [18]. The SH model stems from broader literature on coupled human and natural systems (CHANS) (also called socioecological systems or social-ecological systems) which provide a method of understanding whole systems that have both human and natural components [19]. The CHANS approach allows for the inclusion of multiple human (e.g., economics, demographics, and politics) and natural sectors (e.g., hydrology and ecology). CHANS-related research strategies have been applied to a wide range of water management topics including through SH [20–23]. While CHANS and SH offer an important advance over other frameworks, one shortcoming is that they do not explicitly allow for the comparison of multiple integrated systems across different spatial scales. Due to the increasing global nature of threats to freshwater supplies, it is vital to consider interactions within and across these integrated systems. The metacoupling framework is a powerful new approach that builds on previous CHANS research and methods by allowing for integration across spatial scales (see Table 1 for some comparisons between the metacoupling framework, HEM, and SH).

**Table 1.** Comparison of some features of the metacoupling framework to hydro-economic modeling (HEM) and socio hydrology (SH) for watershed-level research.

|  | Metacoupling | HEM | SH |
|---|:---:|:---:|:---:|
| Accounts for both socioeconomic and environmental characteristics | X | X | X |
| Can predict future scenarios | X | X | X |
| Explicitly integrates across local to regional scales | X |  |  |
| Can account for cultural and spiritual characteristics and governance styles | X |  | X |

CHANS are both individually complex and they have complex relationships with other CHANS across varied spatial and temporal scales [24]. The metacoupling framework is a newly proposed framework for comparing multiple CHANS across a variety of spatial scales from local to global [25]. It explicitly analyzes interactions within and between CHANS across varied distances.

The metacoupling framework has been applied to the study of water transfer projects, urban watersheds, and non-transboundary watersheds, but has yet to be applied to transboundary watershed systems [20,21,26]. Wang et al. highlight the usefulness of the metacoupling framework for integrating social and ecological components in the management of large watersheds within China [27]. Building on this research, we argue that the metacoupling framework is a useful method for studying social and ecological components across multiple integrated spatial scales of transboundary river basins. The ability to integrate across neighboring and distant systems can help improve the management of these vital but threatened systems. To demonstrate this, we present a case study of the transboundary Limpopo River Basin in Southern Africa. This large watershed is home to millions of people and increasing threats are leading the once perennial river to become seasonal in some locations [28,29]. We begin by explaining the metacoupling framework and applying it to the Limpopo River Basin to the extent possible given current limitations in data. We then provide a roadmap with which other researchers can further apply and adapt the metacoupling framework to this and other transboundary watershed systems.

## 2. The Limpopo River Watershed Case Study

The Limpopo River watershed consists of approximately 400,000 km$^2$ or 1.3% of the African continent [28,29]. The watershed supports 18.8 million people living in South Africa, Botswana, Zimbabwe, and Mozambique in addition to 102 dams and thousands of small-scale reservoirs and irrigation projects [30,31] (Figure 1). Water availability in the Limpopo River has declined as demand for water continues to increase throughout the basin; the primary drivers of this increasing demand are population growth and urbanization [32]. The once perennial Limpopo River is now dry for several months in an average year [29]. Water demand in the basin is expected to increase by 46% by 2025 [33]. The watershed has experienced severe droughts alternating with floods over several decades, with 1.2 million people needing immediate food assistance in 2016 after a drought that caused up to 100% crop losses in some areas [33]. Yet agriculture currently accounts for 60% of the watershed's water use [33]. Although mining accounts for a smaller proportion of water use (around 10%), it is expected to increase in the coming years and has caused severe environmental consequences via pollutants [34]. Around 80% of the basin experiences water stress [35]. There is currently a water deficit in the basin whereby 500 million cubic meters of water is transferred from the Orange Senqu River Basin each year [33] to meet demands. Despite such scarcity, transnational level conflicts over water security have not yet occurred at the level as seen in other parts of the world (e.g., the Mekong) [34]. Yet recent transnational efforts have been initiated to allow for increased dialogue across countries, such as The Resilience in the Limpopo River Basin (RESILIM) Program administered by the United States Agency for International Development (USAID) [34].

Within this large watershed we highlight a particular focus on small-scale irrigation projects led by the Limpopo National Park in the southern region of Mozambique. Data for this case study came from internal reports produced by the Limpopo National Park and personal observations during an irrigation scheme assessment in July 2018. The evaluation report contains data on scheme area, number of members, functionality of the pump/pipe system, challenges, benefits, governance styles, and other qualitative data related to the associations.

The Limpopo National Park partnered with a non-governmental organization in 2012 to implement irrigation projects or schemes in 18 communities located in the Game Management Zone on the park boundary and within the Limpopo River water basin (Figure 2). Each community formed an association and they were provided with a pump and pipes to bring water from the nearby river (either the Limpopo River or a tributary) to the association's land. Currently, 16 of these original schemes are still in existence and being monitored regularly by extension staff from the park. In addition to improving household income, livelihoods, and food security, the projects were meant to foster positive relationships between the communities and the park. The size of each association ranges from 11 to 43 members (average = 28.75) and the size of land under irrigation ranges from 2 to 8 ha (average = 5)

(Table 2) [36]. Each member is allocated a portion of the association's land in which to farm individually. Individuals are given access to the pump on a rotation of about 1 or 2 weeks depending on the size of the group. Furrow technique is used to deliver water throughout the fields. Overirrigation is common with the use of furrows, particularly when they are manually opened and closed as in these schemes [5].

**Table 2.** Limpopo National Park irrigation schemes.

| Scheme | Number of Members | Total Area in Hectares |
|---|---|---|
| Chimangue | 40 | 4.3 |
| Hassane | 21 | 2 |
| Matafula | 7 | 2.5 |
| Ndope | 42 | 4 |
| Psitima [1] | 28 | 3 |
| Chibotane | 43 | 6.5 |
| Chipanzo | 29 | 2.5 |
| Cunze | 20 | 4 |
| Guzwe | 12 | 5 |
| Macuachane | 35 | 6 |
| Munhamane | 28 | 7 |
| Chibumba | 31 | 4 |
| Mbeti | 40 | 12 |
| Sihogoni | 20 | 8 |
| Licenga | 11 | 5 |
| Chicumbane | 41 | 4 |
| Panhame | 40 | 4 |

[1] Was not operating in 2018.

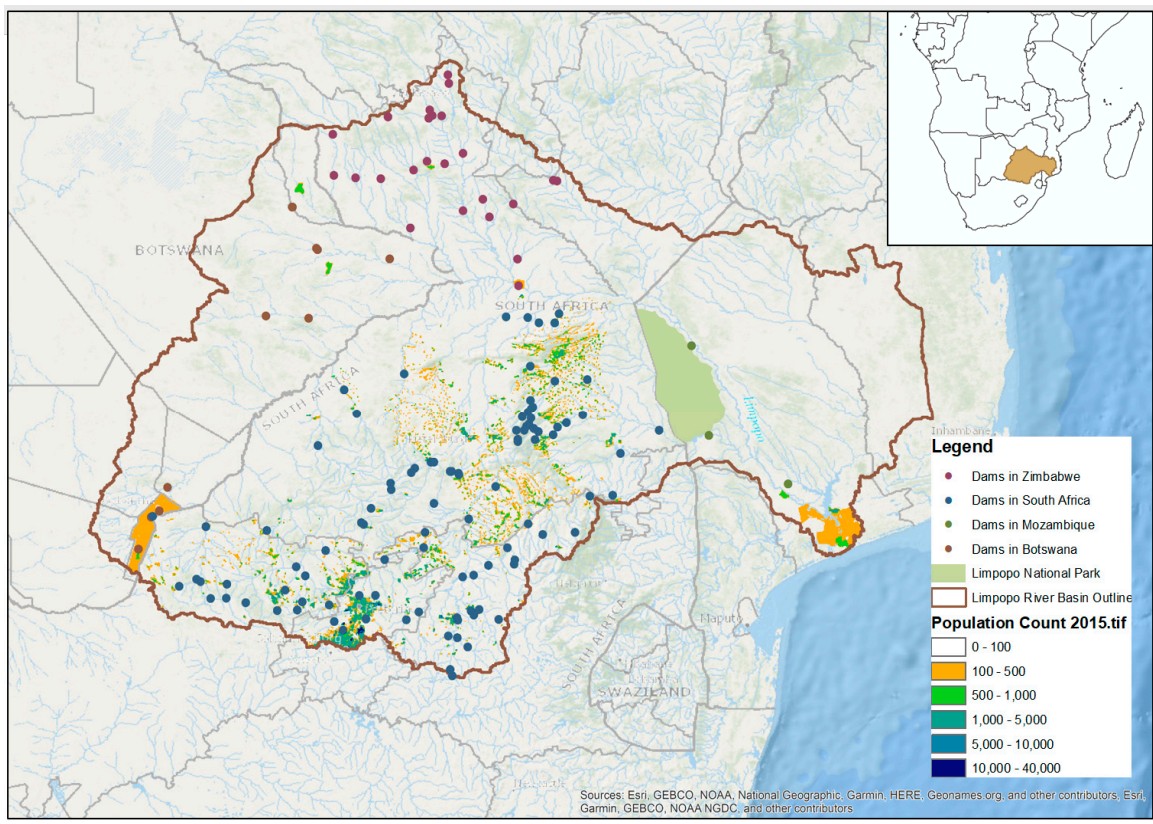

**Figure 1.** Map of the Limpopo River watershed with population density and Limpopo National Park [30,31].

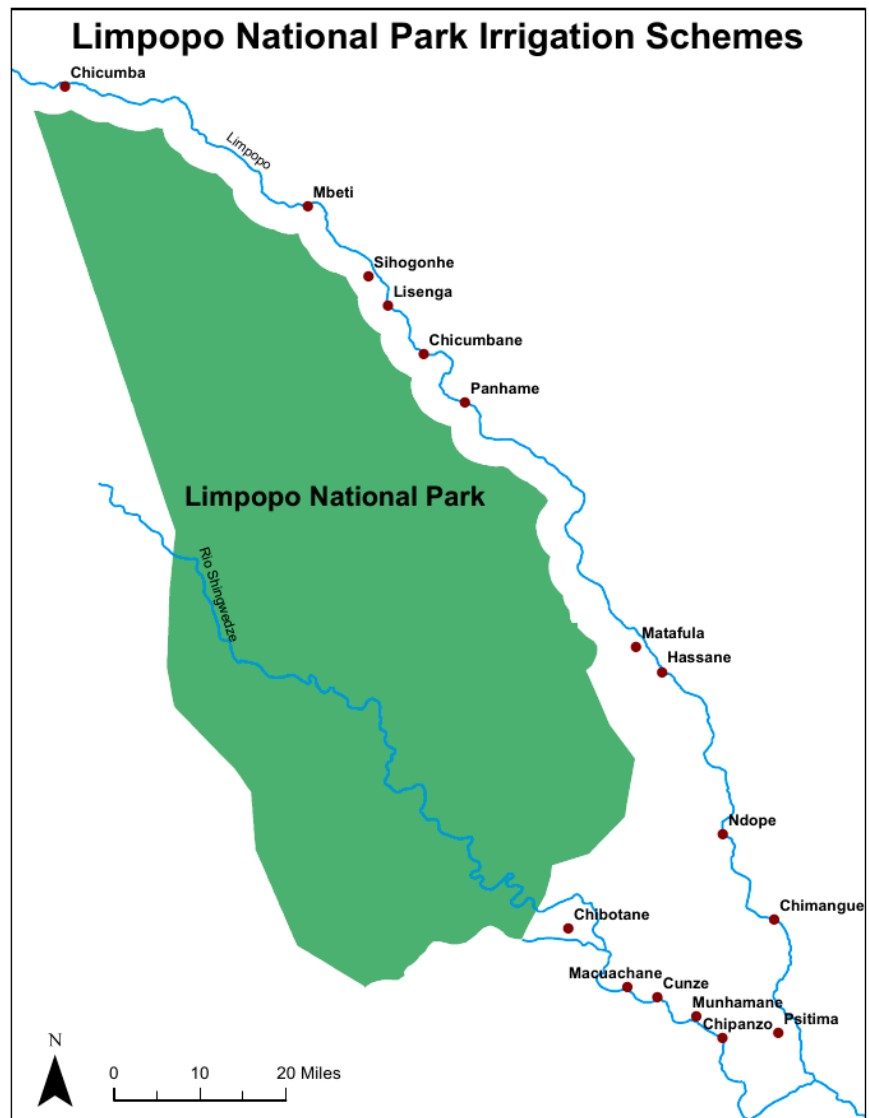

**Figure 2.** Map of the Limpopo National Park irrigation schemes on the Limpopo and Shingwedze Rivers.

## 3. Metacoupling Framework

### 3.1. Overview

Any system in which humans and nature interact can be defined as a coupled human and natural system. While CHANS are individually complex, they also have complex relationships with other CHANS through nested hierarchies [24,37]. Individual systems both affect and are affected by other systems at a variety of scales from local to global [25]. Liu [25] developed the metacoupling framework that defines human–nature interactions within a system (intracoupling), interactions between adjacent systems (pericoupling), and interactions among distant systems (telecoupling). As these definitions are scale-dependent, the distance considered to fall under telecoupling versus pericoupling varies and should be defined according to the system in study. Each system is defined by the behavior of a set of agents who facilitate and respond to causes and effects of the coupling, while also generating a set of flows (of information, energy/matter, or money) that may travel within or among systems. A key topic of interest is how processes at one level affect the other levels and how spillovers or feedbacks may percolate across levels to affect the sustainability of one part of the system or the whole. Telecoupling is becoming increasingly important in the CHANS literature as the prevalence of distant transactions is increasing [38,39] and has been applied to a variety of water-related contexts [20,21,26,40]. The study

of metacoupling is particularly important for understanding global threats like air pollution, climate change, biodiversity loss, food security, and water scarcity [25]. To date, it has been applied broadly to the environment in China [41] and more specifically to soil conservation [42] and large national watersheds [27] but has not yet been applied to examine transnational watershed management.

For the Limpopo River watershed case study, the local irrigation schemes in Limpopo National Park interact with one another via "intracouplings" within the same coupled human and natural system (Figure 3). Interactions among agents at the national level of Mozambique are considered here as "pericouplings" or intermediate level interactive processes. The broadest level includes interactions across the four countries in the watershed—Botswana, South Africa, Zimbabwe, and Mozambique. Interactions across these countries are considered "telecouplings" in this particular system due to the increased distance and the difficulty of sharing information and coordinating resource management across these international boundaries. A summary of the main couplings and cross-scale interactions is shown in Table 3.

**Table 3.** Summary of the application of the metacoupling framework to the Limpopo watershed case study.

| Concept | Description | Effects |
|---|---|---|
| Intracoupling | • Interactions among individual irrigation associations in Limpopo National Park (currently low) | • None seen at this time (low interaction levels) <br> • Potential future water scarcity |
| Pericoupling | • Interactions between Limpopo National Park and national-level entities in Mozambique (e.g., Ministry of Agriculture) | • Fluctuating GDP <br> • Decreasing food security <br> • Water pollution |
| Telecoupling | • Mozambique is telecoupled with South Africa, Botswana, and Zimbabwe (e.g., via national-level agricultural policies) | • Water pollution <br> • Water scarcity <br> • Agricultural land increase |
| Cross-level interactions | • Extension agents facilitate information flows | • Diffusion of information promotes irrigation technology spread <br> • Seeds and harvests also disseminated across space |
| | • Regional climate change promotes local vulnerabilities | • Floods and droughts cause sudden local loss of irrigation not explained by local events |
| | • Transnational politics create cross-level inequalities | • Mozambique is disadvantaged in this system relative to the other countries, resulting in heightened degradation |

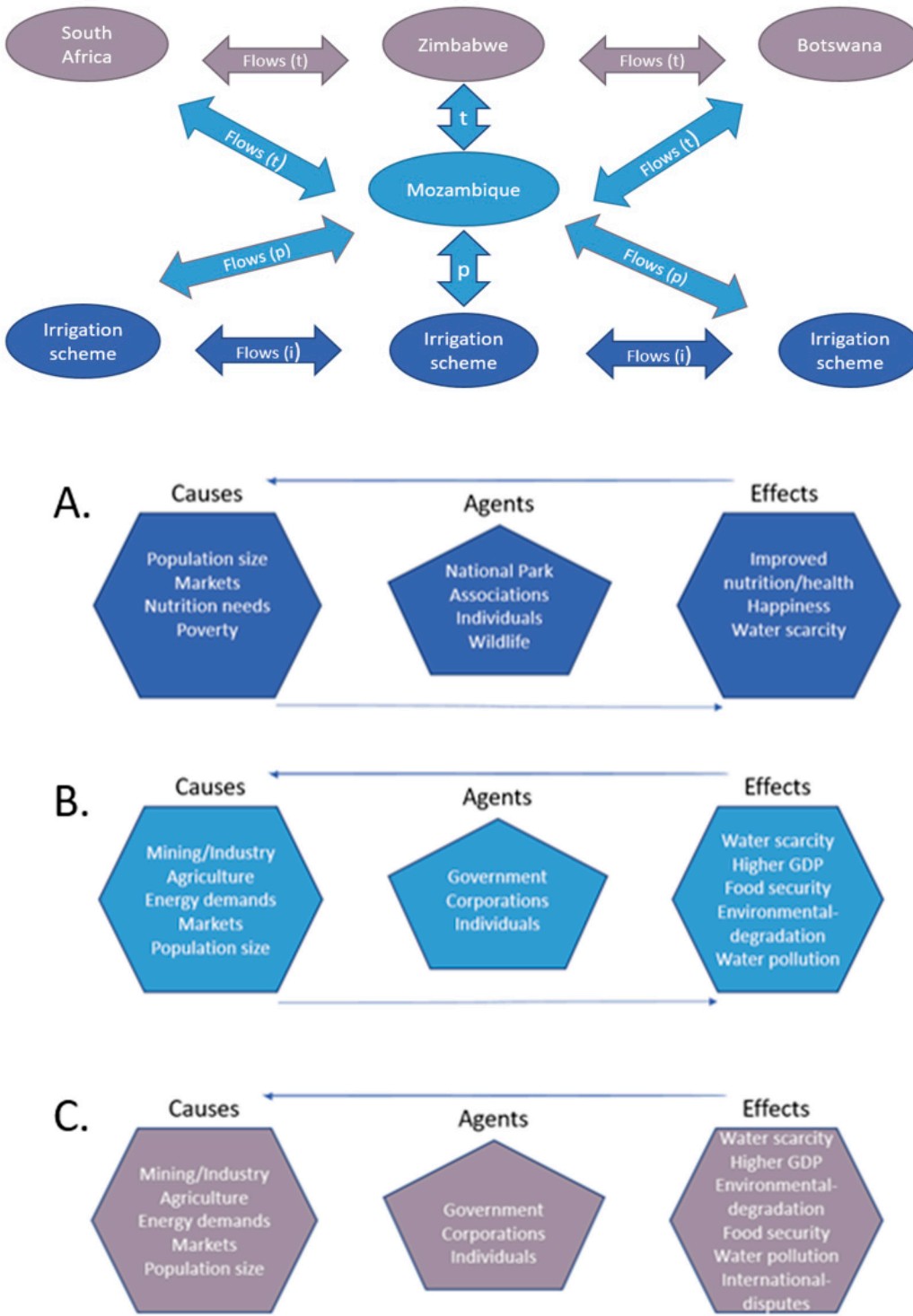

**Figure 3.** Metacoupling framework for the Limpopo River watershed; a general framework showing intracouplings (i), pericouplings (p), and telecouplings (t) with international scale shown in gray, national scale in light blue, and local scale in dark blue. The causes, agents, and effects of systems are shown in (**A**) for local scale, (**B**) for national scale, and (**C**) for regional scale.

*3.2. Intracouplings*

The intracouplings in our case study occur at the local level between individual irrigation schemes implemented around Limpopo National Park. The primary causes of water use are domestic use related to population size, local market demand for agricultural products, nutritional needs, and

poverty (Figure 3A). Population size within 30 km of Limpopo National Park has grown by 20% from 52,837 in 2005 to 62,994 in 2015 [24]. Even if per capita water use remains unchanged, the growing population needs more water for basic domestic uses like drinking, cooking, and washing. Nationally, 54% of Mozambicans live below the poverty line and 80% rely on rain-fed subsistence agriculture [43]. Using basic irrigation systems is one possibility for individuals to improve their food security and potentially increase their income to move above the poverty line. As local agricultural markets are often distant and transportation is unavailable, the project has been reduced to mainly subsistence level food production with some informal local sales. Similarly, inputs (seeds, pesticides, fertilizers, tools, and parts for pump repair) are extremely difficult to obtain and therefore rarely used. The low availability and affordability of seeds in this region has been highlighted in previous studies [28].

The agents are the irrigation associations, individual members, Limpopo National Park, and the wildlife that also rely on the water resource. The level of direct interaction among agents from each irrigation scheme is low as they work independently, but extension agents from the park work across different schemes and thus establish collective norms around irrigation use. As the farms and associated irrigation projects are located next to a national park, damage from wildlife is common with rats being the most destructive species, but other animals like monkeys, hippos, and buffalo also contribute to crop damage.

Improved nutrition, improved health, happiness, and water scarcity are all potential or realized effects of the system. In fact, every association that was in operation in July 2018 reported improved household food security as a result of the irrigation project [36]. Improved health and happiness were also cited as benefits of the project. However, currently the scale of the irrigation projects is small and has not had noticeable effects on water scarcity and quality. In the future, it is possible that the irrigation project can affect water availability and quality at the local level as well as the remaining downstream portion of the watershed. The upstream schemes can contribute to the reduction of water flow by using excessive amounts of water. Additionally, poor agricultural practices can lead to increased runoff of fertilizer or pesticides contributing to water pollution and eutrophication [44]. Given the current size of schemes and the overall lack of inputs, it is currently highly unlikely that there is a noticeable effect of one scheme on another. However, increases in the size or number of schemes could have an impact on overall water scarcity. While none of the schemes are currently using fertilizer and few pesticides are applied, there is potential for this to change. Increases in fertilizer and pesticide use, particularly if not managed properly, could result in increased water pollution in the Limpopo River.

### 3.3. Pericouplings

At the national level, Mozambique is pericoupled to the individual schemes through a variety of government organizations. The CHANS for Mozambique consists of government, corporations, and individuals that use water for mining/industry, agriculture, and energy (Figure 3B). These uses are mainly driven by economic market fluctuations and population growth. The population in Mozambique's portion of the watershed increased by 15% from 1,001,848 in 2005 to 1,153,709 in 2015 [31]. The area under irrigation in Mozambique's portion of the Limpopo River watershed was estimated to be 9400 ha in 2005, with plans to add an additional 77,000 ha by 2024 [45]. This rate of expansion in irrigation will continue to increase water demand drastically.

The effects of water use can be both positive and negative as they include water scarcity, higher GDP (gross domestic product), food security, environmental degradation, and water pollution. All of these effects have been observed in recent years in Mozambique. In 2015, Mozambique reached the Millennium Development Goal of reducing the number of chronically food insecure people by half [46]. GDP has generally been increasing in Mozambique from 4.227 billion in 1997 to 9.367 billion in 2007 and 12.646 billion in 2017 [47]. In terms of environmental impacts, one example is the estimated loss of 1800 ha of mangrove forest per year due to salinization [48]. There is also evidence of increased pollutants in the Limpopo River watershed within Mozambique [32].

The national government interacts with individual schemes through a variety of government organizations across multiple scales [49]. The schemes have regular interactions with Limpopo National Park staff that are ultimately managed by the National Agency of Conservation Areas (Administração Nacional das Áreas de Conservação or ANAC). Each district also has an agricultural extension agent who works for the Ministry of Agriculture. This agent interacts directly with the irrigation associations and many schemes received seeds and advice from the agricultural extension agent. Governance and management styles vary from local to national scales and these differences can also impact availability and quality of water throughout the basin.

*3.4. Telecouplings*

Mozambique is telecoupled with South Africa, Botswana, and Zimbabwe; it receives water and/or pollutants from these upstream countries. As a transnational water basin, each country (Mozambique, Botswana, South Africa, and Zimbabwe) that shares the resource determines national level regulations for water management; these are highly impacted by national politics which can vary both across borders and over time. Agents are government, corporations, and individuals. Multiple water management districts in each country are responsible for managing water availability and quality within their districts. However, there is currently no formal regional level coordination among the different countries that would advise or regulate decisions on how to manage the telecoupled system at the watershed scale.

At the national level and international levels, the countries involved can be considered as individual CHANS. The primary causes of the telecoupling are population size, economic market fluctuations, and demand for mining/industry, agriculture, and energy. Population size throughout the watershed has increased from 15.9 million in 2005 to 18.8 million in 2015 [31]. Agriculture is one of the prime economies in the region and market prices fluctuate widely, but there has been a trend of converting forests, shrublands, and grasslands into agriculture over the previous decade (see Figure 4). According to landcover classifications from ESA-CCI-LC and MERIS GlobCover, the amount of agricultural land within the Limpopo River basin nearly doubled from 160,000 km$^2$ in 2016 to 300,000 km$^2$ in 2019 (see Figure 5) [50,51].

The potential positive and negative effects are decreased water availability, increased water pollution, environmental degradation, higher GDP, improved food security, and energy access. Telecoupling effects are identical for each of the four countries (as represented in Figure 3C) with changes only in the levels/importance of each aspect.

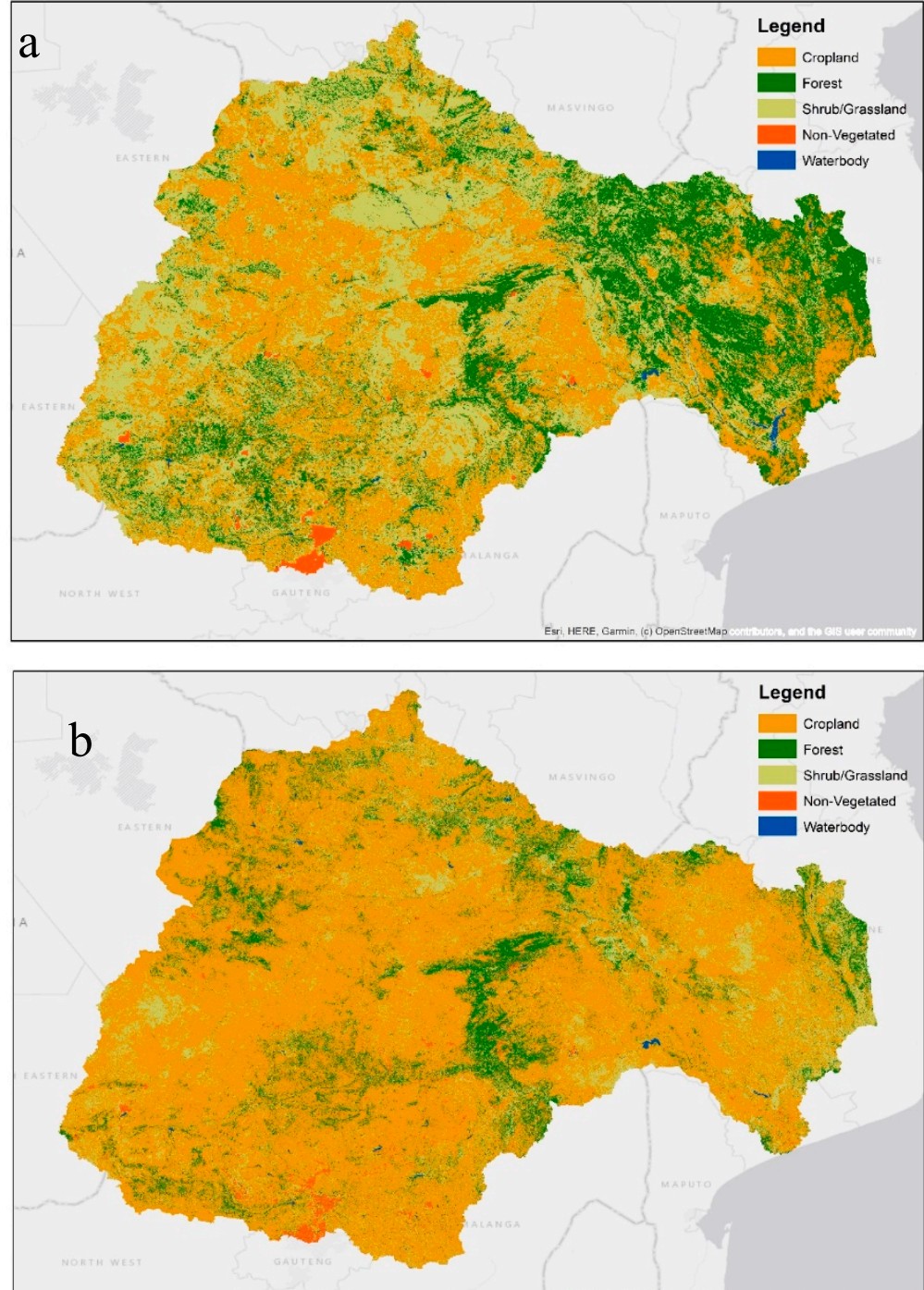

**Figure 4.** Landcover classifications for the Limpopo River Watershed in Southern Africa [50,51]. (**a**): 2009 Landcover classification; (**b**): 2016 Landcover classification.

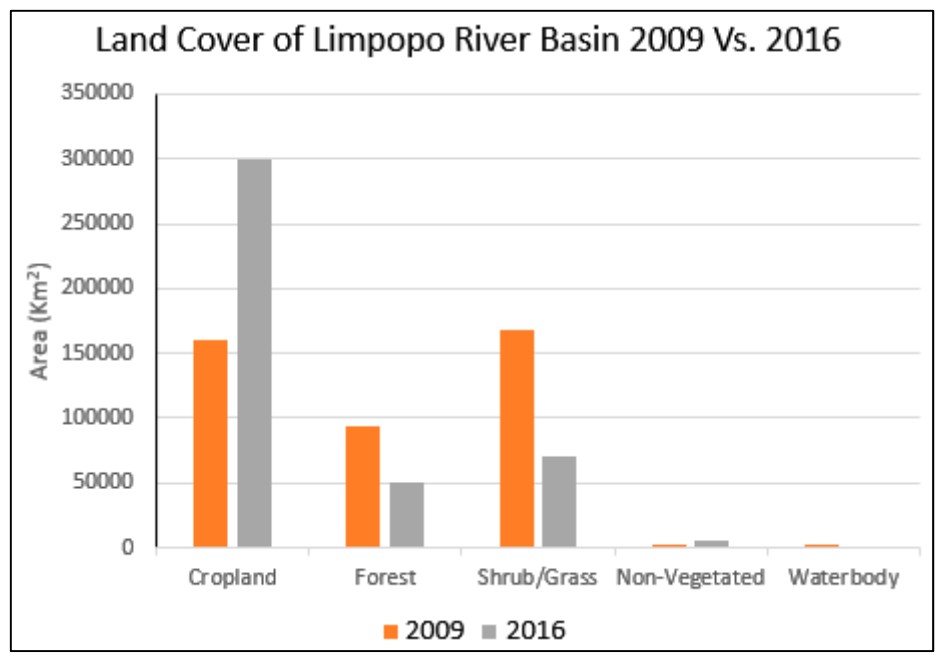

**Figure 5.** Land cover types and quantities in the Limpopo River Basin from 2009 to 2016 [50,51].

## 4. Interactions and Feedbacks among Levels of the Metacoupled System

### 4.1. Extension Agents Facilitate Information Flows within and among Systems

One of the most interesting aspects of our system was observing the powerful mechanism in which flows of information about the irrigation schemes are facilitated within and across systems. At the level of intracouplings, each scheme operates autonomously with little to no direct interaction with the other schemes. The individual schemes have self-organized and determined their own rules and procedures [36]. However, the schemes are relatively close to each other in space and they all receive some oversight and assistance from the park's extension agents. These extension agents provide a means of conveying information between schemes. Any successful agricultural techniques adopted or developed by one scheme will be highlighted to other schemes that can choose to try the newer technique or not. Similarly, extension agents also bring information from outside the schemes as they are trained in agriculture and interact with government agriculture extension agents, generating a flow at the level of broader 'pericouplings' with adjacent systems. Aside from information, these extension agents help facilitate the movement of seeds, inputs, parts, fuel, and/or harvests. As physical markets are distant, and transport is unavailable or unaffordable, the extension agents often purchase goods at the nearest market on behalf of the association members.

### 4.2. Vulnerability to Climate-Induced Floods and Droughts

Another key observation that was apparent in our system is that the success of the local irrigation projects was vulnerable in the face of floods, droughts, and changes in river morphology, forces that were ultimately driven by broader climate patterns beyond the local system. Multiple schemes have been negatively affected by these natural events. Floods occur annually, but if they occur earlier than expected, they can destroy entire fields and damage pumps. Similarly, drought can lead to the loss of an entire harvest. During periods of low water levels in the river, the location of water channels may be too far for the existing pump/pipe set-up to operate. In these cases, several associations have been forced to abandon their land in search of other plots closer to the existing river channels. This resulted in extra land being cleared for agriculture and a significant increase in investment of time and energy and therefore a reduction in the length of the potential growing season. The reason that only 16 of the original 18 schemes are still functional is due to separate severe flood events carrying away

the pumps from the other two schemes [36]. In a 2010 study, Silva et al. [28] found that this region of Mozambique is the most prone to floods and droughts and the most chronically food deprived. Association members have become reliant on these irrigation schemes and even small changes in water availability, quality, or location could potentially have detrimental effects on food security, nutrition, and health within the communities.

### 4.3. Inequality in the Telecoupling Driven by Transnational Politics

We also became acutely aware in our study that the water levels experienced in the local system were limited by broad scale inequality occurring at the transnational level and shaped by broader patterns in economic development. The source of the Limpopo River is in South Africa and it becomes the boundary between South Africa and Botswana as well as between South Africa and Zimbabwe before entering Mozambique. As Mozambique is the farthest downstream of the countries that share the basin, they receive both water and pollutants from the other countries. While the flow of water is a natural process, the countries each choose to allow the natural process to occur by not building dams and/or using more of the water supply for themselves. Furthermore, by opening dams during periods of high rainfall, South Africa can and has sent additional water into Mozambique, resulting in severe floods and destruction of crops [28].

Mozambique is particularly vulnerable to processes occurring outside of its boundaries because 53.8% of the water used comes from a transnational water resource [52]. Limpopo River is one of the primary sources. Mozambique is currently only using approximately 0.68% of their renewable water resources, whereas Botswana, Zimbabwe, and South Africa use 1.59%, 17.85%, and 30.19%, respectively [52]. Botswana uses the lowest proportion of water for agriculture and Zimbabwe uses the highest at over 80% [52]. To date, Mozambique has only exploited a small portion of their renewable water resources and only a small portion of farmland has been converted for irrigation uses. While there is an opportunity to expand water uses to improve agriculture and food security, the potential for this depends greatly on regional factors outside of their control. In a 2010 study, Van der Zaag et al. [45] found that Mozambique has sufficient water availability to expand commercial irrigation in the Limpopo River basin, but development projects in neighboring countries could greatly affect this potential for expansion. As more than half of their renewable water comes from neighboring countries, Mozambique is highly vulnerable to changes in water availability and quality upstream. Dams are one of the primary factors in water quantity, and there are dozens throughout the basin. South Africa has by far the most dams in their portion of the Limpopo River watershed and Mozambique only has three in their portion. The dams in Mozambique provide water for irrigation with only a small percentage being used for hydro-electricity production. South Africa, Botswana, and Zimbabwe have a greater variety of dams that are used for irrigation, water storage, and hydro-electricity production.

## 5. Spillover Effects

Beyond the regional scale, these interactions can have spillover effects globally through pollutants released into the Indian Ocean and changes in the discharge rates of both water and sediment. The Limpopo River discharges 4.8 km$^3$ of water annually into the Indian Ocean. According to Milliman and Farnsworth, the sediment discharge from the Limpopo River decreased by 80% in the past 50 years primarily as a result of dam construction and increased irrigation. The discharge of sediment and organic matter from rivers is important for marine ecosystem health, estuarine flushing, prevention of coastline erosion, and biogeochemical cycles. Spillover effects on the human health and well-being have not been adequately studied and are needed in the future for better conceptualization of this system.

## 6. Conclusions and Future Directions

The management of transboundary watersheds presents a challenge because there are conflicting uses and inequitable distribution of both costs and benefits [12]. The metacoupling framework provides an opportunity to analyze the small-scale management practices within the watershed

while also addressing regional, national, and international relationships that affect management decisions. The decisions made at the local intracoupled scale are both impacted by and can impact water availability and quality in other parts of the basin. Therefore, it is vital that a watershed management plan considers not only the different scales, but also the interactions across scales and causes and effects at each level. The metacoupling framework conceptualized in this paper provides a means of viewing and analyzing all of these different components together for improved management and decision making. This framework can benefit a variety of managers from water districts, irrigation association committees, local governments, regional governments, national governments, international organizations, non-governmental organizations (NGOs), and national parks to work towards sustainable use of the Limpopo River basin.

This paper provides the first attempt to apply the concept of the metacoupling framework to a transboundary watershed, the Limpopo River basin. The case study helps demonstrate the utility of the metacoupling framework and highlights the importance of analyzing transnational watersheds using this approach. In our attempt to apply the novel and valuable framework, we were confronted with a serious lack of publicly available data, particularly on river flow and water use throughout the basin. While this limited our ability to quantify the changes in water availability and quantity throughout the basin, it further highlights the importance of an integrated approach. Using the spatially explicit metacoupling framework would involve the cooperation of multiple agents throughout the basin to collect data in comparable ways and to share these data. This is a vital pre-requisite for using the metacoupling framework to guide water management decisions and can lead to enhanced cooperative management of this transboundary resource that is so vital for the 18.8 million people that rely on this water.

We recommend a follow-up study that provides a more detailed application of the metacoupling framework to the Limpopo River basin. This would require collecting and compiling physical and social data to quantify relationships within and between the various CHANS. Data on river flow and water quality at multiple locations on the river should be collected regularly to identify changes over time and space. Once collected this data should be made publicly available so that all stakeholders can track water quality and availability in the basin. Flow data can be combined with data on precipitation, temperature, and other related variables that are currently available. Similarly, livelihood and ethnographic data on river use and importance should be collected throughout the basin and combined with economic costs of water within the watershed. Social data should be situated within the local management and governance strategies as well as national and international politics. Furthermore, detailed analysis of externalities (e.g., climate change) and spillover effects are needed to better understand the entire system and how it impacts and is impacted by interactions with other types of systems. All relevant data should be combined into a single watershed database that can be accessed by watershed managers and users throughout the four nations to help improve transparency and management. This will require the collaboration of different stakeholders including water managers, traditional leaders, district to national level government officials, non-governmental organization staff, and researchers.

More broadly, the potential for applying metacoupling concepts to transboundary watershed management would benefit from research in four general areas:

1. Applying the metacoupling framework to other transboundary watersheds to assess the generality of its data needs, strengths, and limitations.
2. Exploring the relationships between watersheds and other types of CHANS such as agricultural and mining systems.
3. Investigating the effects of externalities such as climate change on watershed systems.
4. Analyzing the spillover effects from transboundary watershed CHANS (e.g., impacts on biogeochemical cycles, global fisheries).

The integration of a metacoupling framework to transboundary watershed management gives rise to new opportunities and challenges. It allows for the exploration of human–nature interactions within as well as between adjacent and distant systems in transboundary watersheds. This approach complements and goes beyond previous transboundary watershed approaches by integrating across disciplines and scales, providing a transdisciplinary means of analyzing the mounting challenges facing transboundary watersheds in a globalized world.

**Author Contributions:** Conceptualization, L.M. and V.H.; formal analysis, D.Y.; visualization, D.Y. and L.M.; original draft preparation, L.M.; review and editing, V.H., D.Y., and L.M. All authors have read and agreed to the published version of the manuscript.

**Funding:** Leandra Merz was hired as an intern by Peace Parks Foundation to conduct an evaluation of the irrigation scheme project for Limpopo National Park. Peace Parks Foundation provided the funding necessary for the evaluation and the project-related data used in this paper came from the reports generated during this evaluation.

**Acknowledgments:** The authors wish to acknowledge the staff from Limpopo National Park and Peace Parks Foundation that assisted with the irrigation project evaluation in 2018. This includes Peter Leitner, Eric Madamalala, and Oraca Cuambe. Additionally, thank you to Emilio Bruna who provided advice and encouragement during the peer-review process.

**Conflicts of Interest:** The authors declare no conflict of interest.

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
