# Peer review of "A Metacoupling Framework for Exploring Transboundary Watershed Management"

_sustainability, doi:10.3390/su12051879_

Round 1
Reviewer 1 Report
The authors made good improvements and the paper can be accepted as is.
Author Response
Thank you for taking the time to read through all the versions of our paper. We really appreciate your suggestions throughout the process including these positive comments.
Reviewer 2 Report
General comment
The study is interesting and addresses an integrated approach based on metacoupling for transboundary water. It it well written and understandable. However, the novelties should be better presented. Some concerns and suggestions are following:
· In the introduction section, line 48: the word “catchments” maybe can be altered with the word “watersheds”, as in hydrological terminology it describes better larger scale areas, like the ones mentioned herein. A very minor comment.
· Lines 57-69: there are many concerns about which approach is the best or should be followed from the researchers in order to address such integrated problems. HEMs and SH in the latest years, as well as metacoupling are considered usually on the basis of the available data, as you also argue in the manuscript. I believe that the needs of each problem should define the tools, and even their co-operation, thus, I recommend enriching this paragraph with some recent and important literature contributions, in order to provide more info to the future researchers going through these decisions. See the following for HEMs
o Alamanos, A., Latinopoulos, D., Papaioannou, G., & Mylopoulos, N. (2019). Integrated hydro-economic modeling for sustainable water resources management in data-scarce areas. Water Resources Management (2775–2790):1-16. doi: 10.1007/s11269-019-02241-8
o Bekchanov, M., Sood, A. & Jeuland, M. (2015). Review of Hydro-Economic Models to Address River Basin Management Problems: Structure, Applications and Research Gaps. IWMI Working Paper 167, International Water Management Institute, Colombo, Sri Lanka.
o Alamanos, A., Latinopoulos, D., Xenarios, S., Tziatzios, G., Mylopoulos, N. & Loukas, A. (2019). Combining hydro-economic and water quality modeling for optimal management of a degraded watershed. Journal of Hydroinformatics, 21 (6): 1118-1129. doi.org/10.2166/hydro.2019.079.
o Esteve P, Varela-Ortega C, Blanco-Gutiérrez I, Downing TE (2015) A hydro-economic model for the assessment of climate change impacts and adaptation in irrigated agriculture. Ecol Econ 120:49–58. https://doi.org/10.1016/j.ecolecon.2015.09.017
and for SH:
o Sivapalan, M., Savenije, H.H.G. and Blöschl, G. (2012), Socio‐hydrology: A new science of people and water. Hydrol. Process., 26: 1270-1276. doi:10.1002/hyp.8426
o Di Baldassarre, G., Sivapalan, M., Rusca, M., Cudennec, C., Garcia, M., Kreibich, H., et al. (2019). Sociohydrology: Scientific challenges in addressing the sustainable development goals. Water Resources Research, 55, 6327– 6355. https://doi.org/10.1029/2018WR023901
o Di Baldassarre, G., Viglione, A., Carr, G., Kuil, L., Salinas, J. L., and Blöschl, G.: Socio-hydrology: conceptualising human-flood interactions, Hydrol. Earth Syst. Sci., 17, 3295–3303, https://doi.org/10.5194/hess-17-3295-2013, 2013.
· Lines 71-74: Following my previous comment I recommend adding some more references providing definitions of the metacoupling concept.
· Line 86: ideal framework compared to HEM and SH? I’d suggest adding also a comment acknowledging that it is i) case specific, ii) depends on the goals and iii) the available data. I find this statement too general.
· In section 2 I suggest adding a better description of the region’s pressures, resources rivalries, pollution sources, in general the competitive forces, the disagreements, previous agreements among countries and also a short review on previous studies. Thus, you will strengthen the importance of the studied problem and highlighted the novelty of your approach (in an area that has been studied intensively).
· Figure 3: If I am not mistaken, this is the adjustment of the metacoupling concept into the transboundary problem studied. This is also the basis for the following sections. Since it is a new point of view in transboundary water management, I recommend adding a comment explaining its pros and cons. For example, do you believe that there are any factors not included in this integrated scheme? Any externalities? For HEMs or SH could be compared or applied in some of this system’s components? If climate change for example is an externality, HEMs can address it. So it should be clear the point of view that you examine the management problem. Answering questions like these will help you develop a trade-off among metacoupling, HEMs and SH, leading to useful findings. It will be a novel comparison showing differences, similarities and ways of working complementary.
· If possible, a Table to summarize the factors described in Sections 3.1, 3.2, 3.3, 3.4, 4.1, 4.2, and 4.3, (and their components) with a short description or in the form of cause-effect will help the reader leave with a solid picture of this work.
· In the conclusion section, the actions or proposals-suggestion came out of this work should be further highlighted: to decision-makers, scientists of different disciplines, stakeholders (if applicable), etc. in order to show the importance of your contribution.
Author Response
General comment
The study is interesting and addresses an integrated approach based on metacoupling for transboundary water. It it well written and understandable.
Thank you for the positive feedback.
However, the novelties should be better presented. Some concerns and suggestions are following:
- In the introduction section, line 48: the word “catchments” maybe can be altered with the word “watersheds”, as in hydrological terminology it describes better larger scale areas, like the ones mentioned herein. A very minor comment.
Thanks. We have changed catchment to watershed
- Lines 57-69: there are many concerns about which approach is the best or should be followed from the researchers in order to address such integrated problems. HEMs and SH in the latest years, as well as metacoupling are considered usually on the basis of the available data, as you also argue in the manuscript. I believe that the needs of each problem should define the tools, and even their co-operation, thus, I recommend enriching this paragraph with some recent and important literature contributions, in order to provide more info to the future researchers going through these decisions. See the following for HEMs
o Alamanos, A., Latinopoulos, D., Papaioannou, G., & Mylopoulos, N. (2019). Integrated hydro-economic modeling for sustainable water resources management in data-scarce areas. Water Resources Management (2775–2790):1-16. doi: 10.1007/s11269-019-02241-8
o Bekchanov, M., Sood, A. & Jeuland, M. (2015). Review of Hydro-Economic Models to Address River Basin Management Problems: Structure, Applications and Research Gaps. IWMI Working Paper 167, International Water Management Institute, Colombo, Sri Lanka.
o Alamanos, A., Latinopoulos, D., Xenarios, S., Tziatzios, G., Mylopoulos, N. & Loukas, A. (2019). Combining hydro-economic and water quality modeling for optimal management of a degraded watershed. Journal of Hydroinformatics, 21 (6): 1118-1129. doi.org/10.2166/hydro.2019.079.
- Esteve P, Varela-Ortega C, Blanco-Gutiérrez I, Downing TE (2015) A hydro-economic model for the assessment of climate change impacts and adaptation in irrigated agriculture. Ecol Econ 120:49–58. https://doi.org/10.1016/j.ecolecon.2015.09.017
Thank you for bringing this literature to our attention. We now mention and cite Esteve et al., Alamanos et al., and Bekchanov et al. in our paper.
and for SH:
o Sivapalan, M., Savenije, H.H.G. and Blöschl, G. (2012), Socio‐hydrology: A new science of people and water. Hydrol. Process., 26: 1270-1276. doi:10.1002/hyp.8426
o Di Baldassarre, G., Sivapalan, M., Rusca, M., Cudennec, C., Garcia, M., Kreibich, H., et al. (2019). Sociohydrology: Scientific challenges in addressing the sustainable development goals. Water Resources Research, 55, 6327– 6355. https://doi.org/10.1029/2018WR023901
- Di Baldassarre, G., Viglione, A., Carr, G., Kuil, L., Salinas, J. L., and Blöschl, G.: Socio-hydrology: conceptualising human-flood interactions, Hydrol. Earth Syst. Sci., 17, 3295–3303, https://doi.org/10.5194/hess-17-3295-2013, 2013.
Thanks. We now mention and cite Baldassarre et al., and Sivapalan et al. in our paper.
- Lines 71-74: Following my previous comment I recommend adding some more references providing definitions of the metacoupling concept.
Thanks. We have now added Wang et al. and Schafer-Smith et al. as two additional examples of metacoupling.
- Line 86: ideal framework compared to HEM and SH? I’d suggest adding also a comment acknowledging that it is i) case specific, ii) depends on the goals and iii) the available data. I find this statement too general.
Good point. We have changed “ideal” to “useful” and clarified that we think it is particularly useful for analyzing the multiple integrated spatial scales of transboundary systems. We do mention the limitations of previous methods for addressing integration across multiple spatial scales and present the metacoupling framework as a “useful” framework for integrating spatial scales which we believe can improve watershed management. We have added text in paragraph 4 to acknowledge that metacoupling can build on SH and still incorporate the human and natural components of this framework within the hierarchy of spatial scales.
- In section 2 I suggest adding a better description of the region’s pressures, resources rivalries, pollution sources, in general the competitive forces, the disagreements, previous agreements among countries and also a short review on previous studies. Thus, you will strengthen the importance of the studied problem and highlighted the novelty of your approach (in an area that has been studied intensively).
We have now added additional text to address this at line 136.
- Figure 3: If I am not mistaken, this is the adjustment of the metacoupling concept into the transboundary problem studied. This is also the basis for the following sections. Since it is a new point of view in transboundary water management, I recommend adding a comment explaining its pros and cons. For example, do you believe that there are any factors not included in this integrated scheme? Any externalities? For HEMs or SH could be compared or applied in some of this system’s components? If climate change for example is an externality, HEMs can address it. So it should be clear the point of view that you examine the management problem. Answering questions like these will help you develop a trade-off among metacoupling, HEMs and SH, leading to useful findings. It will be a novel comparison showing differences, similarities and ways of working complementary.
We have now added a new table (table 1) to address the differences among these three frameworks, as suggested. We also include text in lines 67-80 that clarify how metacoupling can build on the SH framework.
- If possible, a Table to summarize the factors described in Sections 3.1, 3.2, 3.3, 3.4, 4.1, 4.2, and 4.3, (and their components) with a short description or in the form of cause-effect will help the reader leave with a solid picture of this work.
Good suggestion. We have now added a new table (table 3) as suggested.
- In the conclusion section, the actions or proposals-suggestion came out of this work should be further highlighted: to decision-makers, scientists of different disciplines, stakeholders (if applicable), etc. in order to show the importance of your contribution.
Thanks. We have added lines 404-406 to clarify that the recommendations in lines 390-403 should apply to a variety of stakeholders, including the ones listed here and any effort to follow these recommendations will need to be collaborative.
Thank you for the helpful suggestions that have further improved this manuscript. Length constraints unfortunately prevented us from making even further additions.
Round 2
Reviewer 2 Report
The authors have provided satisfactory responses to the comments. The revised version of the manuscript is significantly improved. The study is a contribution to the metacoupling perspective literature, as it addresses a new perspective of application.
Author Response
Thank you for taking the time to read our manuscript and give us feedback.
This manuscript is a resubmission of an earlier submission. The following is a list of the peer review reports and author responses from that submission.
Round 1
Reviewer 1 Report
General comment
The study is interesting and addresses an integrated watershed management view. However, the article is focused in the social aspect, presenting various data and the connection among them is not clear. My main inquiry is how water managers will make use of these information, since a detailed analysis is needed to supplement this work.
Comments
· The followed approach is very close to the concept of hydro-economic modelling. Could you explain why you do not model the system’s characteristics (e.g. water balance, costs, profits, etc.) or ever working on Equilibrium Models (P.E.M. or G.E.M.), leading possibly to an optimization of the system (your case is more like a global optimum function). It is a question that arises to the reader, so it is very important to add a short comment on these to strengthen your approach’s theoretical background. Thus, your study’s impact and applicability will increase.
· I suggest adding more details about the 1) data you used with their sources, 2) a Table with the total values (or summing up) of the factors used in the couplings (the factors presented in Figure 3 I suppose) for each part of you study area. 3) Pls make clear how the factors are connected, in every coupling relation (equations, assumptions, etc).
· A more detailed description of the irrigation schemes is required.
· How is the political factor involved in your analysis? I suggest making it clear from the beginning, or at least in the telecoupling section (either as an assumption, either as an uncertainty regarding each country’s legislation).
· Some of the countries studied are using a bottom-up, or a top-down management approach, or a community management, or all of them in cases? Following my previous comment, could you please explain the legislation and the situation to avoid these questions arising?
· As you mention “it is vital that a watershed management plan consider not only the different scales, but also the interactions across scales and causes and effects at each level”. If that was the purpose of the study, how did you address it? Could you suggest some measures to improve the system overall?
Minor comments
· In the introduction section, it would be good if you could:
o connect better the mentioned research topic (metacoupling, all the factors involved and explaining briefly how they are connected)
o refer to the existing literature approaches,
o further explain the gaps in the existing literature, and thus
o highlight your research questions, the novelties and scopes of your study.
· I recommend adding a Table with some hydrological elements of Limpopo River watershed, if applicable (e.g. rainfall, ET, runoff, inflows, outflows, etc.). Thus, the conditions of water scarcity will be much more understandable, and the reader will have a clearer picture of the area’s problems.
· The resolution of Figure 3 is too poor to read it.
· Figure 4 has to be reformulated in a more easy-to-read way, according to the journal’s format.
· I recommend adding Tables or diagrams for presenting the findings of sections 4.2 and 4.3.
Author Response
Responses (R) in bold font:
The study is interesting and addresses an integrated watershed management view. However, the article is focused in the social aspect, presenting various data and the connection among them is not clear. My main inquiry is how water managers will make use of these information, since a detailed analysis is needed to supplement this work.
R: We appreciate your comment on the applicability of this work directly to water managers. This paper is written to a general audience that includes but is not limited to water managers. We propose the metacoupling framework as a useful conceptualization for water managers, particularly in transboundary watersheds. We have revised the introduction and conclusion to more clearly describe our aims in presenting this new approach as an effective framework for understanding transboundary watersheds. We continue to use the Limpopo River case study as a means of explaining this new framework and highlighting the importance but clarify that this is a general application of the framework and more detailed analysis is needed to supplement this work and make it more practical to watershed managers.
Comments
The followed approach is very close to the concept of hydro-economic modelling. Could you explain why you do not model the system’s characteristics (e.g. water balance, costs, profits, etc.) or ever working on Equilibrium Models (P.E.M. or G.E.M.), leading possibly to an optimization of the system (your case is more like a global optimum function). It is a question that arises to the reader, so it is very important to add a short comment on these to strengthen your approach’s theoretical background. Thus, your study’s impact and applicability will increase.
R: Thank you for bringing this other approach to our attention. We agree that the hydro-economic modelling is similar in many ways. However, we believe that the metacoupling framework goes beyond the scope of the hydro-economic modelling in allowing for comparisons across multiple scales within and between different coupled human and natural systems. We have added descriptions of two recent approaches to watershed management, hydro-economic and hydro-social modeling in our literature review and state the similarities and differences between these and the metacoupling framework in lines 55-71.
I suggest adding more details about the 1) data you used with their sources, 2) a Table with the total values (or summing up) of the factors used in the couplings (the factors presented in Figure 3 I suppose) for each part of you study area. 3) Pls make clear how the factors are connected, in every coupling relation (equations, assumptions, etc).
R: The purpose of this paper is to present the metacoupling framework in relation to transboundary watersheds. We use the Limpopo River as a case study to explain the framework and highlight its usefulness. However, quantifying relationships is beyond the scope of this paper. We recommend additional analysis for this case study that would include more data and attempt to better quantify relationships. Lines 344-358 in the conclusion address the need for follow-up studies.
A more detailed description of the irrigation schemes is required.
R: We now more clearly describe the irrigation schemes in the case study section of the new version of the paper.
How is the political factor involved in your analysis? I suggest making it clear from the beginning, or at least in the telecoupling section (either as an assumption, either as an uncertainty regarding each country’s legislation).
R: Thank you for bringing up the importance of the political factor. We now mention the importance of international politics in lines 229-232 and recommend consideration of political factors as well as differences in management styles and governance for future research on this case study (see lines 350-352).
Some of the countries studied are using a bottom-up, or a top-down management approach, or a community management, or all of them in cases? Following my previous comment, could you please explain the legislation and the situation to avoid these questions arising?
R: We now highlight the importance of management and governance in lines 225-226 and recommend this be included in future work on metacoupling in the Limpopo River watershed case study.
As you mention “it is vital that a watershed management plan consider not only the different scales, but also the interactions across scales and causes and effects at each level”. If that was the purpose of the study, how did you address it? Could you suggest some measures to improve the system overall?
R: Good point. We have added recommendations in the conclusion section to address how this case study could be improved as well as how the metacoupling framework could continue to be applied to other transboundary watersheds to more explicitly address interactions across different scales and different types of CHANS.
Minor comments
In the introduction section, it would be good if you could:
connect better the mentioned research topic (metacoupling, all the factors involved and explaining briefly how they are connected)
R: The introduction has largely been re-written. We more clearly introduce metacoupling.
refer to the existing literature approaches,
R: Existing literature has now been more clearly summarized in the introduction section.
further explain the gaps in the existing literature, and thus
R: Gaps have now been enunciated (see lines 61-63, 68-71, 77-80, 82-84).
highlight your research questions, the novelties and scopes of your study.
R: Following the enunciation of gaps above, we have described the purpose of the paper in lines 88-91
I recommend adding a Table with some hydrological elements of Limpopo River watershed, if applicable (e.g. rainfall, ET, runoff, inflows, outflows, etc.). Thus, the conditions of water scarcity will be much more understandable, and the reader will have a clearer picture of the area’s problems.
R: We appreciate this suggestion but did not have enough data to construct such a table. However, we have included this in the directions for future research section in the conclusion.
The resolution of Figure 3 is too poor to read it.
R: This figure has been updated with better resolution.
Figure 4 has to be reformulated in a more easy-to-read way, according to the journal’s format.
R: This figure has been removed.
I recommend adding Tables or diagrams for presenting the findings of sections 4.2 and 4.3.
R: A large portion of these sections were removed from the paper.
Reviewer 2 Report
It is an interesting document that shows all the "possible" interractions among water users in a very important river of Africa. In my opinion, this article is just a report showing the interrelations among users (which is quite complex, I could say) but the scientific soundness of the article is very low, since there are not new insights or approaches given. The title gave me the impression that a serious analysis regarding irrigation schemes would follow, providing some data at least but the overall discussion was lost inside socioeconomic terms. The only relevant text appears in lines 121-134 and Table 2. This paper should provide more that FAOSTAT data, GDPs and polulations. The legend and the elements of Figure 1 are not readable, Figure 3 is not readable. Data of figure 4 show statistics of the whole countries and for the case study area. Fig.4 has elements which are not in full description.
Author Response
Responses (R) in bold font
It is an interesting document that shows all the "possible" interactions among water users in a very important river of Africa. In my opinion, this article is just a report showing the interrelations among users (which is quite complex, I could say) but the scientific soundness of the article is very low, since there are not new insights or approaches given.
R: Thank you for taking time to read our paper, we appreciate this feedback. We have now largely rewritten the introduction and conclusion sections to emphasize the potential contribution of telecoupling to watershed management. The case study has been shortened (due to lack of data). But we now more clearly highlight the new insights and approaches in our work.
The title gave me the impression that a serious analysis regarding irrigation schemes would follow, providing some data at least but the overall discussion was lost inside socioeconomic terms. The only relevant text appears in lines 121-134 and Table 2.
R: The title has been changed to more accurately reflect the text.
This paper should provide more that FAOSTAT data, GDPs and populations.
R: Good suggestion. We did not find sufficient local scale data to complement the national level data that are available in these databases. Therefore, we shifted the paper to have a more conceptual focus.
The legend and the elements of Figure 1 are not readable
R: This figure has been removed.
Figure 3 is not readable.
R: The resolution of this figure has been improved
Data of figure 4 show statistics of the whole countries and for the case study area. Fig.4 has elements which are not in full description.
R: This figure has been removed.
Reviewer 3 Report
The manuscript presents routine work and the findings were not comprehensively discussed. The authors try to use a metacoupling approach for watershed management, while the metacoupling framework is not present clearly. This paper in its present form does not bring new relevant information to the field. There should be some solid theoretical evidence provided to support your proposed idea which should lead to some scientifically significant results.
Introduction should be rewritten. It should be expanded to include a more detailed discussion of current problems, and potential applications as well as the concept of the framework.
Would you explicitly specify the novelty of your work? What progress against the most recent state-of-the-art similar studies was made?
The literature review section should be improved. It should be dedicated to present critical analysis of state-of-the-art related work to justify the objective of the study. Also, critical comments should be made on the results of the cited works.
The discussion statements are speculations. Make every attempt to improve the discussion by critically analyzing your findings.
Conclusions should be amended to incorporate a broader discussion of the significance and potential application of this specific study.
Author Response
Responses (R) in bold font
The manuscript presents routine work and the findings were not comprehensively discussed. The authors try to use a metacoupling approach for watershed management, while the metacoupling framework is not present clearly. This paper in its present form does not bring new relevant information to the field. There should be some solid theoretical evidence provided to support your proposed idea which should lead to some scientifically significant results.
R: Thank you for taking the time to read our paper, we appreciate the helpful feedback. We have re-focused the paper to be more conceptual with the case study used to explain and highlight the relevance of a metacoupling framework in this context. We have now more comprehensively introduced telecoupling and outlined its potential contributions to watershed management. We now present more theoretical evidence and direction for future research.
Introduction should be rewritten. It should be expanded to include a more detailed discussion of current problems, and potential applications as well as the concept of the framework.
R: The Introduction has been largely rewritten to better set the stage for the work. We also more clearly describe the framework itself (see lines 73-91).
Would you explicitly specify the novelty of your work? What progress against the most recent state-of-the-art similar studies was made?
R: We address the novelty of metacoupling in general in lines 73-91 and of the application to transboundary watersheds in lines 75-78 and 82-89. We present the potential value of the framework by comparing it to existing frameworks in lines 61-64 and 68-71 and add recommendations for future work in the conclusion.
The literature review section should be improved. It should be dedicated to present critical analysis of state-of-the-art related work to justify the objective of the study. Also, critical comments should be made on the results of the cited works.
R: We have significantly improved the literature review to include major works in watershed management in recent years in the realm of socio-ecological research and cross-scale approaches.
The discussion statements are speculations. Make every attempt to improve the discussion by critically analyzing your findings.
R: The discussion section has largely been re-written. The case study is now significantly shortened to avoid speculation.
Conclusions should be amended to incorporate a broader discussion of the significance and potential application of this specific study.
R: The conclusion section is now broader, as suggested, and includes directions for future research.
Round 2
Reviewer 1 Report
The authors have made an attempt to address most of the comments. However, I still have some doubts regarding the usefulness of the relations, and if it is better not to present them.
Maybe the title could change to something more theoretical instead of presenting it as metacoupling approach.
Reviewer 2 Report
The authors significantly improved their manuscript.
Reviewer 3 Report
The authors modify the manuscript somehow, while my basic comments are not well stressed. I keep my first decision.